# The effect of using light emitting diodes and fluorescent lamps as different light sources in growth inhibition tests of green alga, diatom, and cyanobacteria

**Akira Okamoto*, Miyuki Imamura, Kazune Tani, Takeru Matsumoto**

Department of Environmental Science & Toxicology, Nippon Soda Co., Ltd., Odawara, Kanagawa, Japan

* a.okamoto@nippon-soda.co.jp

**Data Availability Statement:** All relevant data are within the paper.

## Abstract

Aquatic organisms have been used to investigate the safety of chemicals worldwide. One such assessment is an algal growth inhibition test. Algal growth inhibition tests are commonly performed using a growth chamber with fluorescent lamps as the lighting source, as test guidelines require continuous uniform fluorescent illumination. However, fluorescent lamps contain mercury, which has been identified as hazardous to humans and other organisms. The Minamata Convention (adopted in 2013) requires reduction or prohibition of products containing mercury. On the other hand, light-emitting diodes do not contain mercury and provide a photosynthetically effective wavelength range of 400–700 nm which is an adequate light intensity for algal growth. Light-emitting diodes are thus preferable to fluorescent lamps as a potential light source in algal growth inhibition tests. In this study, we investigated if light-emitting diodes could be substituted for fluorescent lamps in growth inhibition studies with green alga (*Pseudokirchneriella subcapitata*), diatom (*Navicula pelliculosa*), and cyanobacteria (*Anabaena flos-aquae*). Algal growth inhibition tests were performed using five different chemicals known to have different modes of action and are assigned as reference substances in the test guidelines. The results of each algal test showed similar values between light-emitting diodes and fluorescent lamps in terms of conditions for the growth inhibition rate and percent inhibition in yield of each chemical. It was therefore concluded that using light-emitting diodes instead of fluorescent lamps as a lighting source had no effect on the algal growth inhibition test results.

## Introduction

Owing to certain advantages over fluorescent lamps (FLs) and incandescent lamps (conventional lighting source), light-emitting diodes (LEDs) are increasingly being used for devices such as mobile phones, cameras, televisions, outdoor billboards, cars, trains, airplanes, ships, interior lights, and external lights [1–3]. LEDs have a range of advantages such as a high response time, wider range of controllable color temperatures, a wider operating temperature range, no low-temperature startup problems, high energy efficiency, low maintenance cost,

**Funding:** Nippon Soda provided the salaries of all authors. Nippon Soda provided equipment and facilities, but played no role in study design, data collection and analysis, decision to publish, or preparation of the manuscript. The specific roles of all authors are articulated in the 'author contributions' section.

**Competing interests:** Nippon Soda provided the salaries of all authors. There are no patents, products in development or marketed products to declare. This does not alter our adherence to PLOS ONE policies on sharing data and materials.

and long life [1]. Furthermore, FLs are one of the mercury-containing products that the Minamata Convention (adopted in 2013) is required to reduce or prohibit, as mercury is hazardous to humans and the environment.

Despite the initial cost of LEDs being higher than that of FLs, LED usage has seen a recent increase because LEDs do not contain mercury and have a low health impact as a result of low ultraviolet radiation (UV) [1, 4]. LEDs have also been used for illumination in greenhouse plants. Singh D et al. (2015) said "It is necessary to further investigate the not yet fully understood physiological processes mediating plant responses to LED light; however, economic analysis has clearly shown that LEDs can reduce the electricity cost, and investment (high capital cost) will be returned as profit in long-term operations in greenhouse industries" [5]. Reports about the effect of LEDs on plant growth have led to several studies comparing LEDs with FLs.

Numerous aquatic organisms have been used for safety investigation tests of chemicals worldwide, one of which is the algal growth inhibition test with continuous illumination. Guidelines for algal growth inhibition tests have been published by public agencies such as the Organization for Economic Cooperation and Development (OECD), United States Environmental Protection Agency, and the Japanese Chemical Substances Control Act (JCSCA). Currently, algae in algal growth inhibition tests are commonly cultured using growth chambers with FLs. These tests are in compliance with guidelines that state that "the surface where the cultures are incubated should receive continuous, uniform fluorescent illumination" [6–9]. One article showed "phosphor-converted (pc)-LEDs and FLs have similar emission spectra, so pc-LEDs may be suitable as an FL replacement within the same color temperature range without causing significant changes in algal growth rates and biochemical properties" [10]. However, this does not necessarily prove that LEDs are suited for FLs replacement, as no comparison test between LEDs and FLs has been conducted.

Light is essential for the growth of algae, and light sources are therefore important for algal growth inhibition tests. In this study, we conducted algal growth inhibition tests on green alga (*Pseudokirchneriella subcapitata*), diatom (*Navicula pelliculosa*), and cyanobacteria *(Anabaena flos-aquae)*. We used similar types of incubator shakers with LED and FL conditions to evaluate whether LEDs can be used as a substitute for FLs. We selected *Pseudokirchneriella subcapitata* (*P. subcapitata*), *Navicula pelliculosa* (*N. pelliculosa*), and *Anabaena flos-aquae* (*A. flos-aquae*) because these algae are used to evaluate the ecotoxicity of industrial chemicals and pesticides and are recommended in test guidelines [6–8]. We conducted tests for five chemicals, namely, NaCl, $K_2Cr_2O_7$, $CdCl_2$, 3,5-dichlorophenol (DCP), and pentachlorophenol (PCP) because these chemicals are assigned as reference substances in test guidelines and have different modes of action (MOA). The MOA of each test chemical simulates osmotic stress (NaCl), photosynthesis inhibition ($K_2Cr_2O_7$), oxidative stress ($CdCl_2$), respiration inhibition (3,5-DCP), and oxidative phosphorylation uncoupling (PCP) [11–16]. If LEDs meet the illumination requirements of test guidelines, the results of the tests between the LED and FL are deemed not to be different, regardless of the difference in the MOA of the chemicals.

## Materials and methods

### Algal culture

*P. subcapitata*, *N. pelliculosa*, and *A. flos-aquae* are recommended in OECD test guideline No. 201 and OCSPP number 850.4500 and 850.4550 [6–8], and the test methods are internationally recognized. Moreover, the results of these species are required in registrations of pesticides or industrial chemicals. Therefore, these three algal species were used for this study.

Axenic cultures of *P. subcapitata* and *A. flos-aquae* were obtained from American Type Culture Collection (ATCC), Manassas, VA, USA. These strains were identified as ATCC

22662 and ATCC 29413, respectively. The cultures were kept at 23°C ± 1°C, under continuous illumination using a white FL in an OECD test medium. Meanwhile, an axenic culture of *N. pelliculosa* was obtained from the UTEX Culture Collection of Algae, Austin, TX, USA. The strain is UTEX661. The culture was kept at 23°C ± 1°C, under continuous illumination using a white FL in an AO-NP test medium (original medium). The original medium is based on an AAP medium, as indicated by the OECD guidelines [6]. The components of the AO-NP test medium are AAP medium ×20 and HEPES. However, $FeCl_3$ was changed to $FeSO_4$, and the volume of Fe ions is not ×20 but ×1.

## Substances and test solutions

Test substances were purchased from FUJIFILM Wako Pure Chemical Corporation, Ltd. These are reference substances stipulated in the test guidelines, which are used for checking the strain sensitivity and the test procedure of the test facility. The test substances were NaCl (Lot No. DSH1669), $K_2Cr_2O_7$ (Lot No. MCQ7043), $CdCl_2$ (Lot No. WDN6675), 3,5-DCP (Lot No. WKL1643), and PCP (Lot No. 161–08301). NaCl has been determined as a reference substance in the Whole Effluent Toxicity (WET) test in Japan. $K_2Cr_2O_7$ is a reference substance in terms of JCSCA and the "Data Requirements for Supporting Registration of Pesticides" (JMAFF) [17] as well as the WET test. $CdCl_2$ is a reference substance of the International Organization for Standardization (ISO) 8692 [18] and the United States Environmental Protection Agency method 1003.0 [19]. 3,5-DCP is a reference substance of the OECD test guideline No. 201 (OECD TG 201), ISO 8692, and WET. PCP is a reference substance of JMAFF.

NaCl, $K_2Cr_2O_7$, and $CdCl_2$ were directly dissolved in a test medium to prepare each test solution. To prepare each stock solution, 3,5-DCP and PCP were dissolved in *N*,*N*-dimethylformamide (DMF). These stock solutions were each added into test medium for each test solution preparation. DMF was used as the solvent, and the final DMF concentration was 0.1 mL/L in each test solution, as per OECD TG201.

## Incubator shakers

The incubator shakers used were PRA35-R-L (PRECI Co., Ltd.) with LED and MR-100L (Takasaki Scientific Instruments Corp.) with FL. The LED (PRA35-R-L) type was EL-L01-LT103F-DQM-D2418 (electric consumption: 18 W, Tsujiko Co., Ltd), whereas the FL (MR-100L) type was FLR20S•W/M•A (electric consumption: 20 W, Toshiba Lighting & Technology Co.).

## Growth inhibition test conditions

Each algal growth inhibition test was conducted as per OECD TG201. The tests were performed using both LED and FL incubator shakers at the same time for each substance. Each test consisted of six replicates for the untreated control (or solvent control) and three replicates for each substance with their corresponding concentrations. The dilution factor between each concentration was 2 (geometric series). The test vessel used was a 300-mL glass Erlenmeyer flask containing 100 mL of the test solution. Each flask was sterilized and covered with an aluminum cap to avoid contamination and evaporation but allowed for gas exchange. Light intensities of *P. subcapitata* and *N. pelliculosa* growth inhibition tests were found to be within 60–80 $\mu E/m^2/s$, whereas that of the *A. flos-aquae* growth inhibition test was within 50–60 $\mu E/m^2/s$. Initial cell densities of *P. subcapitata*, *N. pelliculosa*, and *A. flos-aquae* were 5,000, 10,000, and 7,000 cells/mL, respectively. The test temperatures were 23°C ± 1°C for *P. subcapitata* and 24°C ± 1°C for *N. pelliculosa* and *A. flos-aquae*. All algal tests were conducted with shaking (100 rpm).

## Observation and measurement

The temperature and light intensity were measured daily in each incubator shaker. Light intensities in the incubator shakers were measured at five points using an illuminometer (LI-1400; MEIWAFOSIS Co., Ltd.) in each test. The pH was measured at the initiation and termination stages of each test. The algal cells were inspected at the termination of each test using a microscope (BX51 TRF; Olympus Corp.). The cell densities of *P. subcapitata* and *N. pelliculosa* were measured using an electronic particle counter (CDA-1000; Sysmex Corp.), whereas that of *A. flos-aquae* was measured using a microplate reader (Infinite 200; Tecan Japan Co., Ltd.). Measurements were taken at 24, 48, and 72 h after the initiation of each test. Spectral distributions and chromaticity diagrams at 70 $\mu E/m^2/s$ were determined at the center of each incubator shaker using a spectroradiometer (USR-45D; Ushio Inc.), and the absorption peaks of each organism were measured using a spectrophotometer (UV-2250).

## Statistical analyses

The growth inhibition rates of 10%, 20%, and 50% ($E_rC_{10}$, $E_rC_{20}$, and $E_rC_{50}$) and the inhibition in yield of 10%, 20%, and 50% ($E_yC_{10}$, $E_yC_{20}$, and $E_yC_{50}$) were calculated by plotting the log nominal concentration versus growth inhibition rate and percent inhibition in yield using probit analysis, and these were analyzed using the JMP program (SAS Institute Inc.). The No Observed Effect Concentrations (NOEC) of growth inhibition rate ($NOEC_r$) and inhibition in yield ($NOEC_y$) were defined as the highest concentration tested that did not cause any significant growth reduction compared with the untreated control (or solvent control). NOECs were

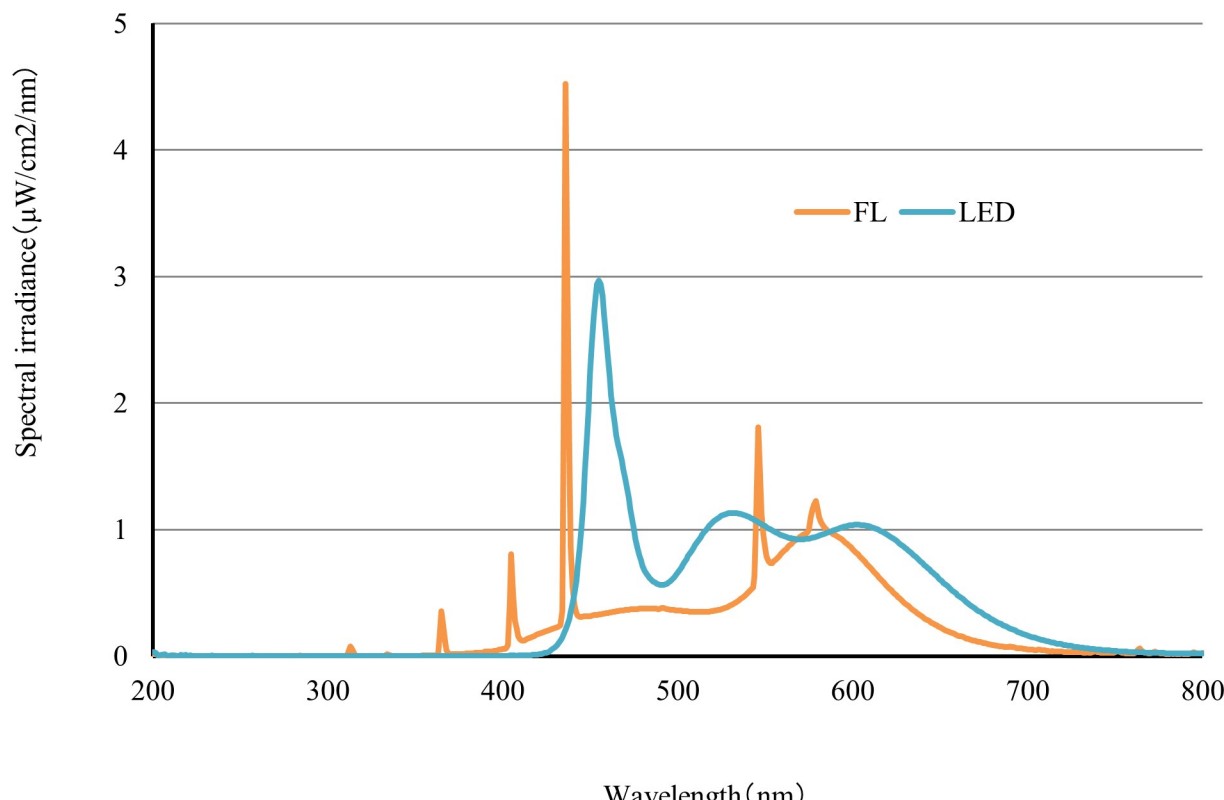

**Fig 1. Spectral distributions of the FL and LED conditions.**

calculated using Williams' multiple comparison test ($\alpha$ = 0.05, one-sided) or Dunnett's test ($\alpha$ = 0.05, one-sided). Williams' test was conducted using the Pharmaco Basic program (Scientist Press Co., Japan) and Dunnett's test using the JMP program.

## Results

### Spectral distributions and chromaticity diagrams

The spectral distributions of the LED and FL conditions are presented in Fig 1. The wavelengths of maximum intensity of the LED and FL conditions were found to be 455 nm and 436 nm, respectively, and their spectral irradiances were 2.97 $\mu$W/cm$^2$/nm and 4.52 $\mu$W/cm$^2$/nm, respectively. The spectral irradiance of the LED condition was 1.0 $\mu$W/cm$^2$/nm or more at 445–475 nm. In addition, the spectral irradiance of the FL condition was 1.0 $\mu$W/cm$^2$/nm or more at 435–438 nm. The wavelength shape of the LED condition was broad, whereas that of the FL condition was sharp; consequently, the shapes of the detected wavelengths and the spectral irradiances were different between the LED and FL conditions. However, the wavelengths of both illuminations included 420–470 nm and 660–680 nm, which are required for the growth of algae [11]. The chromaticity coordinates of the LED condition were x = 0.3153 and y = 0.3197, which was on the slightly blue side of the white point. The chromaticity coordinates of the FL condition were x = 0.3711 and y = 0.3699, which was on the slightly yellow side of the white point. The color temperatures of the LED and FL conditions were approximately 6400 K and 4200 K, respectively.

### Specific light absorption peak for algae

The specific absorption peak for each alga is shown in Fig 2. All algae have chlorophyll *a*, and their necessary wavelengths were determined to be 420–470 nm and 660–680 nm. Moreover,

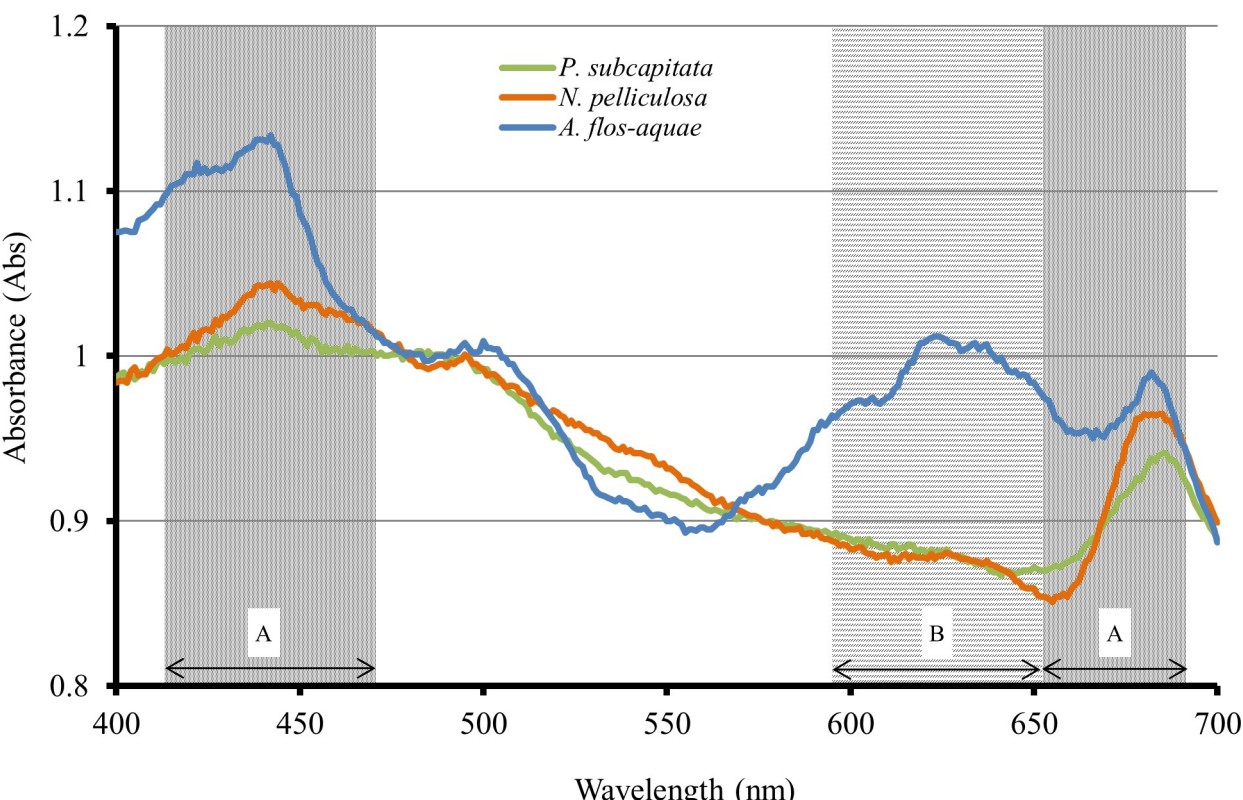

**Fig 2. Absorption peaks of each alga.** A: Chlorophyll *a*, B: Phycocyanin.

*A. flos-aquae* had absorption peaks at 600–650 nm, which could be attributed to the phycocyanins they contain.

## Growth inhibition tests

The growth inhibition rates and percent inhibition in the yield of each concentration in each test are shown in Tables 1–3. The growth inhibition rates and the percent inhibition in yield of each concentration in each test are shown in Figs 3 and 4, respectively. The growth inhibition rates and the percent inhibition in yield of each concentration in each test were determined to be similar between the LED and FL conditions.

## NOEC_r, NOEC_y, E_rC_x, and E_yC_x

$NOEC_r$, $NOEC_y$, $E_rC_x$, and $E_yC_x$ in each test are shown in Table 4. Most endpoints in each test corresponded to the LED and FL conditions. $NOEC_r$ and $NOEC_y$ of a number of tests showed a difference between the LED and FL conditions; however, the $E_rC_{10}$ and $E_rC_{20}$ and $E_yC_{10}$ and $E_yC_{20}$ of these tests corresponded closely. Several $E_rC_{10}$, $E_rC_{20}$, and $E_rC_{50}$ and $E_yC_{10}$ and $E_yC_{20}$ were identified as extrapolation values, as these values were outside the range of test concentrations and calculated from regression analysis.

**Table 1. Growth inhibition rates and percent inhibition in yield of *P. subcapitata*.**

| Compound | Nominal Concentration | Growth inhibition rate (%) | | Inhibition of yield (%) | |
|---|---|---|---|---|---|
| | (mg/L) | LED | FL | LED | FL |
| NaCl | 375 | -3.68 | -2.76 | -23.1 | -17.4 |
| | 750 | 0.616 | 3.14 | 3.80 | 17.2 |
| | 1500 | 14.6 | 13.5 | 56.5 | 54.7 |
| | 3000 | 75.3 | 71.6 | 99.0 | 98.8 |
| | 6000 | 91.7 | 89.9 | 99.8 | 99.8 |
| $K_2Cr_2O_7$ | 0.250 | 4.65 | 3.19 | 24.3 | 16.9 |
| | 0.500 | 20.6 | 14.0 | 70.7 | 55.8 |
| | 1.00 | 60.9 | 51.9 | 97.5 | 95.3 |
| | 2.00 | 86.9 | 85.8 | 99.7 | 99.6 |
| | 4.00 | 93.8 | 92.0 | 99.9 | 99.8 |
| $CdCl_2$ | 0.0125 | 1.50 | 1.73 | 8.65 | 9.82 |
| | 0.0250 | 2.76 | 3.17 | 15.4 | 17.6 |
| | 0.0500 | 9.81 | 9.59 | 44.3 | 43.9 |
| | 0.100 | 36.3 | 33.2 | 88.8 | 85.9 |
| | 0.200 | 72.1 | 72.3 | 98.9 | 98.7 |
| 3,5-DCP | 0.313 | 1.40 | 1.10 | 8.38 | 7.73 |
| | 0.625 | 1.50 | 1.60 | 8.62 | 10.0 |
| | 1.25 | 3.18 | 5.19 | 18.2 | 29.2 |
| | 2.50 | 17.2 | 19.2 | 66.6 | 70.7 |
| | 5.00 | 55.9 | 55.1 | 97.3 | 97.2 |
| PCP | 0.0625 | 0.833 | 0.487 | 6.36 | 10.1 |
| | 0.125 | 10.8 | 7.31 | 49.4 | 41.0 |
| | 0.250 | 77.2 | 64.9 | 99.3 | 98.5 |
| | 0.500 | 84.0 | 82.3 | 99.6 | 99.6 |
| | 1.00 | 77.5 | 72.9 | 99.3 | 99.1 |

**Table 2. Growth inhibition rates and percent inhibition in yield of *N. pelliculosa*.**

| Compound | Nominal Concentration | Growth inhibition rate (%) | | Inhibition of yield (%) | |
|---|---|---|---|---|---|
| | (mg/L) | LED | FL | LED | FL |
| NaCl | 750 | -11.9 | -3.56 | -47.6 | -13.3 |
| | 1500 | 0.181 | 3.54 | -2.22 | 11.5 |
| | 3000 | 11.4 | 5.21 | 31.4 | 15.4 |
| | 6000 | 69.1 | 46.8 | 92.7 | 82.1 |
| | 12000 | 106 | 103 | 101 | 100 |
| $K_2Cr_2O_7$ | 0.038 | 0.594 | -1.27 | 4.03 | -4.00 |
| | 0.122 | 0.344 | -3.37 | 2.28 | -11.8 |
| | 0.391 | 21.4 | 19.4 | 52.0 | 49.1 |
| | 1.25 | 70.2 | 56.7 | 93.5 | 88.1 |
| | 4.00 | 93.6 | 82 | 99.1 | 97 |
| $CdCl_2$ | 1.25 | 4.60 | 4.58 | 15.8 | 13.7 |
| | 2.50 | 9.24 | -1.32 | 18.1 | -2.50 |
| | 5.00 | 46.1 | 39.3 | 78.9 | 73.2 |
| | 10.0 | 104 | 107 | 101 | 101 |
| | 20.0 | 107 | 107 | 101 | 101 |
| 3,5-DCP | 0.156 | -0.317 | -2.80 | -0.8 | -11.8 |
| | 0.313 | -1.31 | 6.03 | -4.3 | 21.4 |
| | 0.63 | -1.05 | 11.4 | -4.1 | 37.0 |
| | 1.25 | 68.8 | 54.7 | 93.3 | 90.1 |
| | 2.50 | 109 | 97.1 | 101 | 100 |
| PCP | 0.000763 | 5.00 | 4.99 | 13.1 | 15.2 |
| | 0.00244 | 13.4 | -4.59 | 32.7 | -16.3 |
| | 0.00781 | 17.0 | 14.5 | 40.0 | 36.2 |
| | 0.0250 | 30.3 | 19.7 | 60.7 | 45.3 |
| | 0.0800 | 67.0 | 51.7 | 89.9 | 81.3 |

## Discussion

Philipp Mayer et al. (1998) showed that the toxicity response of green algae is affected by light intensity [20]. Additionally, it has been shown that the dry weights of plants (radish and spinach) are affected if the light wavelengths required for growth are not included [21]. We therefore conducted algal growth inhibition tests under the same light intensity (approximately 70 µE/m$^2$/s for *P. subcapitata* and *N. pelliculosa* and 50 µE/m$^2$/s for *A. flos-aquae*) under LED and FL conditions. Moreover, we measured the spectral distribution for the LED and FL conditions and investigated whether these affected the test results. Consequently, the spectral distributions of the LED and FL conditions were found to be different, but the growth curves of each concentration were similar between the LED and FL conditions in each test. Even if the spectral distributions of the LED and FL conditions were not equal, the results were almost the same when the light intensities and the wavelengths of 420–470 nm and 660–680 nm required for growth of algae were included.

One article showed that "pc-LEDs and FLs have similar emission spectra, so pc-LEDs may be well suited for FL replacement within the same color temperature range without causing significant changes in algal growth rates and biochemical properties" [10]. The color temperatures of the LEDs and FLs in the present study were approximately 6400 K and 4200 K, respectively. However, even if the color temperatures of the LEDs and FLs were not within the same

**Table 3. Growth inhibition rates and percent inhibition in yield of *A. flos-aquae*.**

| Compound | Nominal Concentration | Growth inhibition rate (%) | | Inhibition of yield (%) | |
|---|---|---|---|---|---|
| | (mg/L) | LED | FL | LED | FL |
| NaCl | 750 | 2.08 | 1.82 | 10.6 | 9.74 |
| | 1500 | 3.08 | 3.72 | 15.2 | 18.8 |
| | 3000 | 5.11 | 7.1 | 23.8 | 32.6 |
| | 6000 | 18.9 | 12.1 | 63.6 | 48.7 |
| | 12000 | 93.4 | 97.7 | 99.6 | 99.9 |
| $K_2Cr_2O_7$ | 0.038 | 4.00 | 2.36 | 18.4 | 12.2 |
| | 0.122 | 41.0 | 9.32 | 87.9 | 39.2 |
| | 0.391 | 81.7 | 67.8 | 99.0 | 97.6 |
| | 1.25 | 108 | 99.0 | 100 | 100 |
| | 4.00 | 107 | 104 | 100 | 100 |
| $CdCl_2$ | 1.25 | 0.32 | 1.53 | 1.79 | 7.95 |
| | 2.50 | 0.00 | 1.13 | 0.105 | 5.79 |
| | 5.00 | 2.92 | 8.74 | 14.3 | 27.4 |
| | 10.0 | 10.9 | 16.7 | 43.4 | 52.0 |
| | 20.0 | 37.2 | 42.5 | 86.0 | 88.9 |
| 3,5-DCP | 0.156 | -0.213 | 1.80 | -1.1 | 9.44 |
| | 0.313 | 1.18 | 5.82 | 5.97 | 26.4 |
| | 0.63 | 4.38 | 6.55 | 20.1 | 29.1 |
| | 1.25 | 82.7 | 83.1 | 99.0 | 99.4 |
| | 2.50 | 110 | 113 | 100 | 101 |
| PCP | 0.875 | 5.75 | 5.62 | 25.9 | 25.2 |
| | 1.75 | 14.5 | 12.8 | 52.9 | 48.1 |
| | 3.50 | 50.8 | 39.4 | 93.2 | 86.7 |
| | 7.00 | 119 | 115 | 100 | 100 |
| | 14.0 | 115 | 117 | 100 | 100 |

color temperature range, the results seem to be the same when the necessary wavelength is included for algal growth. Therefore, LEDs containing the necessary wavelength for growth may be used as a substitute for FLs as a lighting source for algal growth inhibition tests.

The specific absorption peaks of *P. subcapitata*, *N. pelliculosa*, and *A. flos-aquae* were different. However, the results of each algal growth inhibition test showed no difference between the LED and FL conditions. Therefore, using LEDs instead of FLs as a lighting source does not affect the results of algal growth inhibition tests.

In the present study, we conducted algal growth inhibition tests using five kinds of chemicals with different MOAs. It is known that chromium toxicity leads to decreased active reaction of PSII in wheat plants [22]. The sensitivity of *P. subcapitata* cultured in our laboratory has been periodically assessed using $K_2Cr_2O_7$, and the acceptable range of the 72-hour $E_rC_{50}$ values in our laboratory ranged from 0.85–1.3 mg/L (mean min to max; n = 17; tested over November 2010–July 2018; illumination, FL). The $E_rC_{50}$ of $K_2Cr_2O_7$ was 0.90 mg/L in the LED condition, which was within the acceptable range in our laboratory. Moreover, the MOA of the other chemicals also showed the same results for the LED and FL conditions. When the LED condition met the illumination requirements of the test guidelines, the results of the tests under both the LED and FL conditions were almost equal in spite of the different MOAs of the test chemicals.

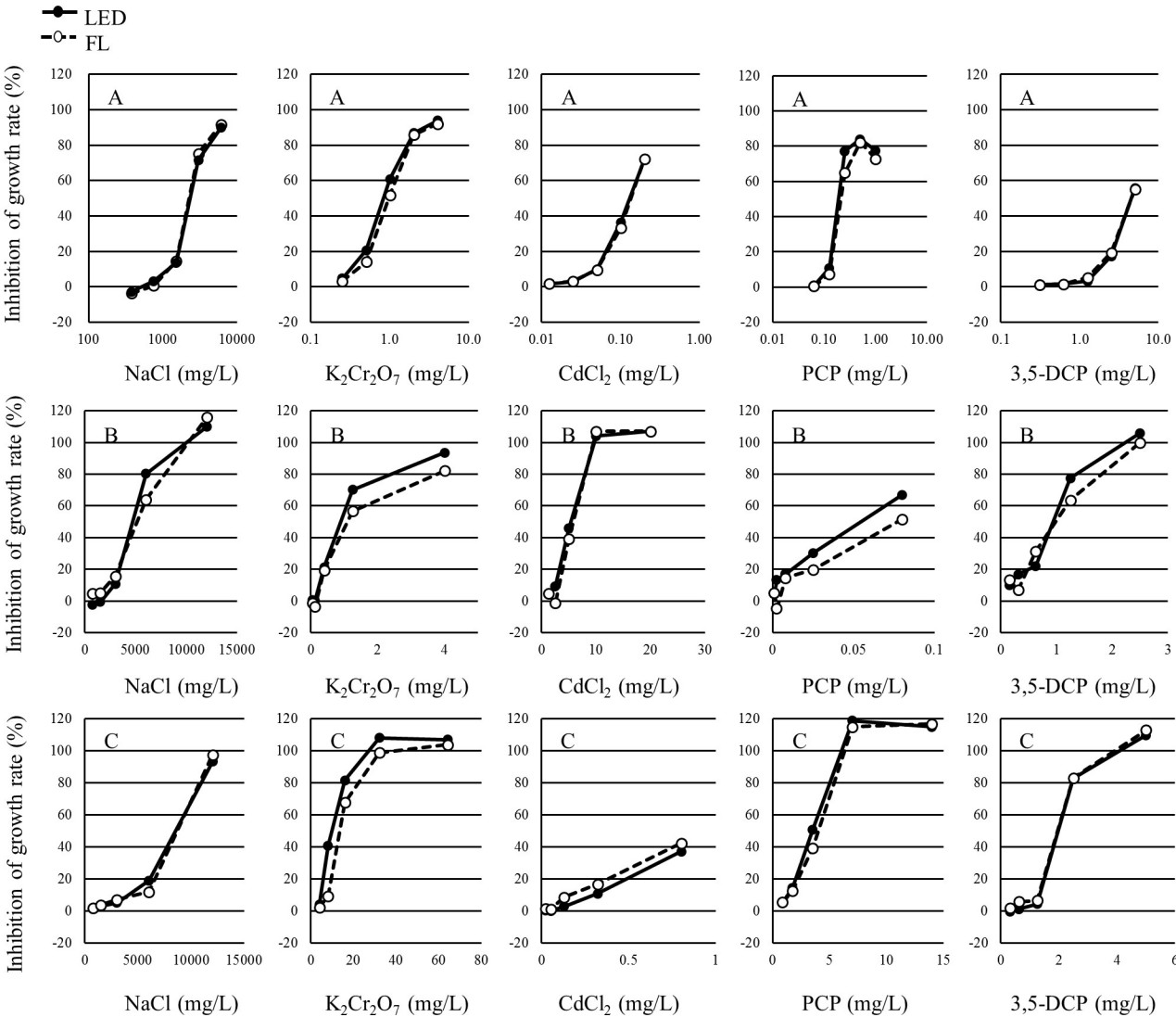

**Fig 3. Concentration-response curve of each alga for the growth rate in each study.** A: *P. subcapitata*, B: *N. pelliculosa*, C: *A. flos-aquae*.

Therefore, if the illumination of the incubator shaker changes from FL to LED, the test results will remain unaffected, as the chemicals are stable against the UV that is contained in the FL. In the present study, we did not use substances with low UV-stability. If the algal growth inhibition test is conducted on substances with low UV-stability under FL conditions, the concentration of this substance is expected to decrease and metabolites will be produced. Whether the results are due to the effects of the test substances or metabolites remains to be determined. Therefore, LED conditions were considered more accurate to evaluate the test substances than the FL conditions. Moreover, LEDs are being used increasingly in the commercial field where it is shown that the initial high capital cost of LEDs is returned as profit in certain sectors due to numerous advantages over FLs. Furthermore, LEDs do not contain mercury and have a low health impact due to low UV. LEDs have benefits in terms of decreased electricity cost and the reduced negative impact on the health of environmental organisms (including humans). In addition, they do not differ from conventional lighting sources when

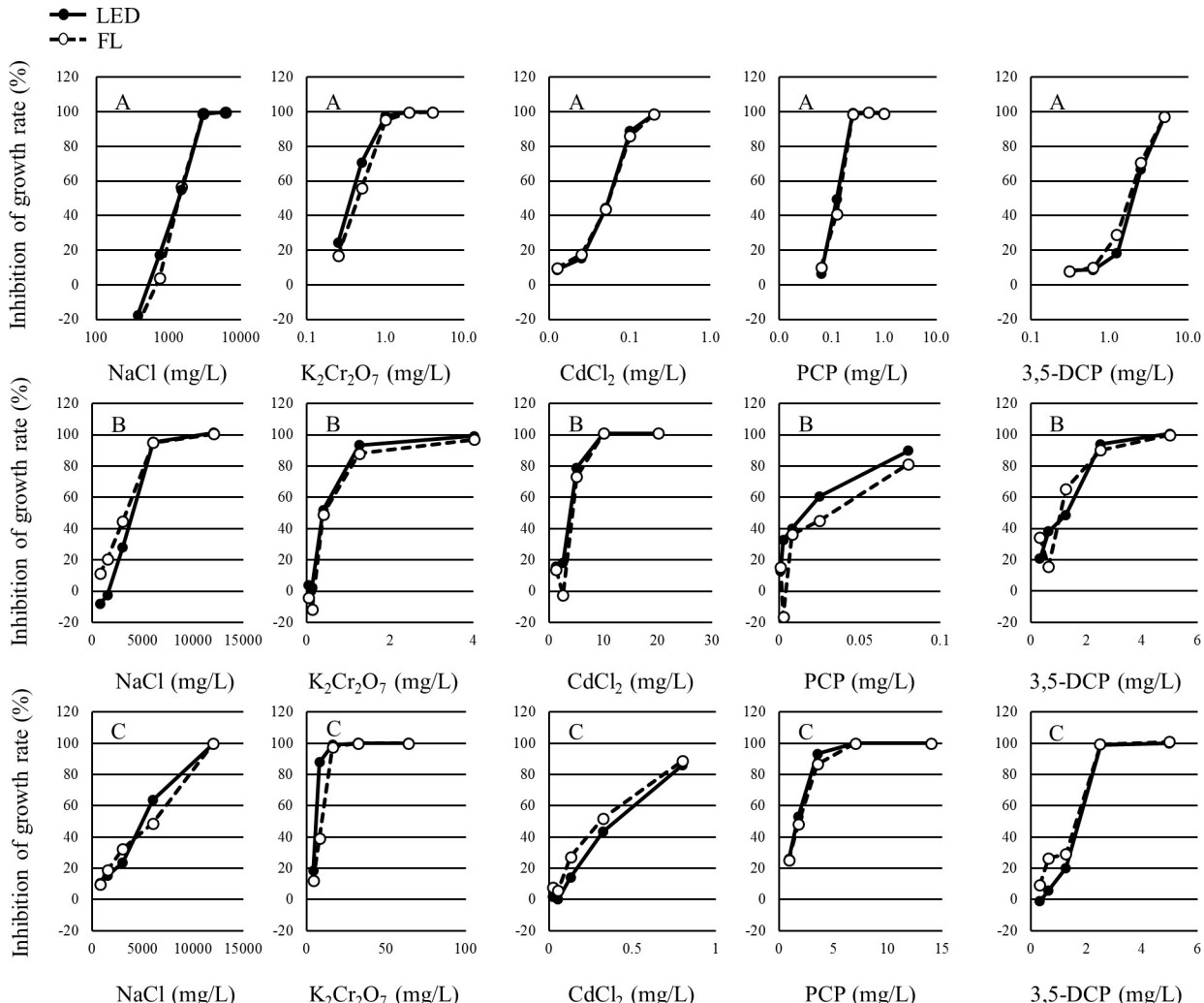

**Fig 4. Concentration-response curve of each alga for the yield in each study.** A: *P. subcapitata*, B: *N. pelliculosa*, C: *A. flos-aquae*.

promoting growth of plants (including algae). LEDs thus appear advantageous over FLs in algal growth inhibition tests. In conclusion, using LEDs instead of FLs as a lighting source did not affect the results of the growth inhibition tests on green alga, diatom, and cyanobacteria. The test guidelines (OECD, OCSPP, and JCSCA) may need to be revised to recommend using LEDs as a lighting source in algal inhibition tests.

## Conclusion

The specific light absorption peaks of *P. subcapitata*, *N. pelliculosa*, and *A. flos-aquae* were found to be different. However, the results of each algal growth inhibition test showed no difference between LED and FL conditions. Moreover, the $E_rC_x$, $E_yC_x$, $NOEC_r$, and $NOEC_y$ values of each chemical under the LED condition showed similar values to those of the FL condition. The use of LED instead of FL as a lighting source did not affect the results of the growth inhibition tests with *P. subcapitata*, *N. pelliculosa*, and *A. flos-aquae*. Therefore, the test guidelines may need to be revised to allow the use of LEDs as a lighting source in algal inhibition tests.

**Table 4. $NOEC_r$, $NOEC_y$, $E_rC_x$, and $E_yC_x$ values of each alga in each test.**

| Algae | Compound | Illumination | $NOEC_r$ (mg/L) | $E_rC_{10}$ (mg/L) | $E_rC_{20}$ (mg/L) | $E_rC_{50}$ (mg/L) | $NOEC_y$ (mg/L) | $E_yC_{10}$ (mg/L) | $E_yC_{20}$ (mg/L) | $E_yC_{50}$ (mg/L) |
|---|---|---|---|---|---|---|---|---|---|---|
| *P. subcapitata* | NaCl | LED | 750 | 1300 | 1700 | 2400 | 750 | 880 | 1100 | 1400 |
|  |  | FL | 750 | 1300 | 1700 | 2500 | 750 | 620 | 840 | 1300 |
|  | $K_2Cr_2O_7$ | LED | <0.25 | 0.34 | 0.48 | 0.9 | <0.25 | 0.19 | 0.23 | 0.37 |
|  |  | FL | <0.25 | 0.41 | 0.57 | 1.0 | <0.25 | 0.20 | 0.27 | 0.44 |
|  | $CdCl_2$ | LED | 0.025 | 0.051 | 0.073 | 0.13 | 0.025 | 0.016 | 0.029 | 0.047 |
|  |  | FL | 0.025 | 0.052 | 0.076 | 0.13 | 0.050 | 0.014 | 0.027 | 0.047 |
|  | 3,5-DCP | LED | 0.625 | 2.1 | 2.6 | 5.0 | 0.625 | 0.69 | 1.3 | 1.7 |
|  |  | FL | 0.313 | 1.8 | 2.6 | 4.9 | 0.625 | 0.52 | 0.97 | 1.6 |
|  | PCP | LED | 0.0625 | 0.089 | 0.11 | 0.26 | 0.0625 | 0.078 | 0.096 | 0.12 |
|  |  | FL | 0.0625 | 0.14 | 0.16 | 0.31 | 0.0625 | 0.062 | 0.095 | 0.13 |
| *N. pelliculosa* | NaCl | LED | 1500 | 2600 | 3100 | 4700 | 1500 | 1800 | 2200 | 3300 |
|  |  | FL | 3000 | 3000 | 3800 | 5700 | 1500 | 1900 | 2500 | 3900 |
|  | $K_2Cr_2O_7$ | LED | 0.122 | 0.23 | 0.35 | 0.81 | 0.122 | 0.1 | 0.16 | 0.36 |
|  |  | FL | 0.122 | 0.27 | 0.45 | 1.2 | 0.122 | 0.17 | 0.24 | 0.49 |
|  | $CdCl_2$ | LED | 1.25 | 2.3 | 2.9 | 4.5 | 2.5 | 1.4 | 1.9 | 3.1 |
|  |  | FL | 2.5 | 2.7 | 3.3 | 5.0 | 2.500 | 1.7 | 2.2 | 3.7 |
|  | 3,5-DCP | LED | 0.625 | 0.67 | 0.79 | 1.1 | 0.625 | 0.49 | 0.59 | 0.84 |
|  |  | FL | 0.313 | 0.52 | 0.66 | 1.0 | 0.313 | 0.29 | 0.37 | 0.63 |
|  | PCP | LED | 0.000763 | 0.0020 | 0.0060 | 0.050 | 0.000763 | 0.00033 | 0.00099 | 0.0082 |
|  |  | FL | 0.00244 | 0.0037 | 0.012 | 0.12 | 0.00244 | 0.00057 | 0.0018 | 0.017 |
| *A. flos-aquae* | NaCl | LED | < 750 | 3100 | 4100 | 7200 | 750 | 1200 | 1800 | 3800 |
|  |  | FL | 750 | 3100 | 4200 | 7200 | 3000 | 1100 | 1700 | 4000 |
|  | $K_2Cr_2O_7$ | LED | < 4.00 | 4.6 | 5.6 | 9.4 | < 4.00 | 3.8 | 4 | 5.5 |
|  |  | FL | < 4.00 | 8.2 | 10 | 13.0 | 4 | 4.30 | 5.3 | 7.9 |
|  | $CdCl_2$ | LED | 0.051 | 0.3 | 0.47 | 1.3 | 0.051 | 0.091 | 0.14 | 0.33 |
|  |  | FL | 0.051 | 0.18 | 0.37 | 1.2 | 0.051 | 0.048 | 0.073 | 0.25 |
|  | 3,5-DCP | LED | 0.625 | 1.2 | 1.4 | 1.9 | 0.625 | 0.8 | 1.2 | 1.4 |
|  |  | FL | 0.313 | 0.91 | 1.1 | 1.7 | 0.313 | 0.43 | 0.6 | 1.1 |
|  | PCP | LED | < 0.875 | 1.4 | 2.1 | 3.0 | < 0.875 | 0.62 | 0.85 | 1.5 |
|  |  | FL | < 0.875 | 1.5 | 2 | 3.2 | < 0.875 | 0.64 | 0.87 | 1.6 |

## Acknowledgments

We thank A. Koizumi (Nisso Chemical Analysis Service Co., Ltd., JP) for the experimental support.

## Author Contributions

**Conceptualization:** Akira Okamoto, Takeru Matsumoto.

**Data curation:** Akira Okamoto.

**Formal analysis:** Akira Okamoto.

**Investigation:** Akira Okamoto, Miyuki Imamura, Kazune Tani.

**Resources:** Akira Okamoto.

**Supervision:** Takeru Matsumoto.

**Writing – original draft:** Akira Okamoto.

**Writing – review & editing:** Akira Okamoto, Takeru Matsumoto.

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
