## [Decision Letter · Decision Letter 0]

3 Nov 2020

PONE-D-20-28966

Effects of different lighting sources on the growth inhibition tests of green alga, diatom, and cyanobacteria

PLOS ONE

Dear Dr. Okamoto,

Thank you for submitting your manuscript to PLOS ONE. After careful consideration, we feel that it has merit but does not fully meet PLOS ONE’s publication criteria as it currently stands. Therefore, we invite you to submit a revised version of the manuscript that addresses the points raised during the review process.

We look forward to receiving your revised manuscript.

Kind regards,

Christophe Hano

Academic Editor

PLOS ONE

Journal Requirements:

"The authors received no specific funding for this work."

We note that one or more of the authors are employed by a commercial company: "Nippon Soda Co., Ltd.,"

Reviewers' comments:

Reviewer's Responses to Questions

**Comments to the Author**

1. Is the manuscript technically sound, and do the data support the conclusions?

Reviewer #1: Yes

Reviewer #2: Partly

2. Has the statistical analysis been performed appropriately and rigorously? 

Reviewer #1: Yes

Reviewer #2: Yes

3. Have the authors made all data underlying the findings in their manuscript fully available?

Reviewer #1: Yes

Reviewer #2: Yes

4. Is the manuscript presented in an intelligible fashion and written in standard English?

Reviewer #1: No

Reviewer #2: No

5. Review Comments to the Author

Reviewer #1: Recommendation: Minor concerns

The manuscript entitled, “Effects of different lighting sources on the growth inhibition tests of green alga, diatom, and cyanobacteria” represents very interesting results, however, there are some minor concern regarding this study. If these concerns/queries are satisfied, it can be considered for publication.

Research title:

•The title of research “effects of different lighting sources” is very general, rephrase it.

In abstract section:

•Please add an introductory line about why algal growth inhibition test is performed, discuss rationale behind the research.

In introduction section:

•Line 44, 45, please elaborate if LED or FL is better? This literature does not support or negate the research study.

•Line 51, “algal growth inhibition tests” are cultured or algae are cultured?

•Line 57-58, “as no comparison test between LED and FL was conducted” please rephrase it with “has been conducted”.

•Please mention about the role of LED/FL illumination in algal growth inhibition tests.

In methodology and results section:

•This section is fine, and methodology and results are in coherence with each other.

•However, the figures added could be of better resolution.

In discussion section:

•Please improve the discussion section, and make your research work more attractive.

•Line 213, this information is not present in the cited reference 10. Please, check.

References:

•The references are not uniformly cited, e.g. reference 10, journal name is abbreviated while rest are not.

Reviewer #2: Comment and suggestion to authors:

PONE-D-20-28966

Titled: "Effects of different lighting sources on the growth inhibition tests of green alga, diatom, and cyanobacteria".

1)In this manuscript, the authors compared only between fluorescent lamp and light-emitting diode as the lighting sources, so it may not suitable to use the title “Effects of different lighting sources…..”. It would be better to improve the title to be more specific and clearer.

2)This study employed only 1 species for green alga, 1 species for diatom and 1 species for cyanobacteria, why did each species can be the representative species to evaluate the effect of different lighting sources on the growth inhibition? This point should be clarified in the manuscript.

3)Why did Pseudokirchneriella subcapitata, Navicula pelliculosa, and Anabaena flos-aquae is suitable for this study? Why don’t the authors used other species or more number of species for their experiment? This point should be clarified in the manuscript as well.

4)There are some spelling mistakes and grammatical error found in this manuscript, the author should pay more attention on this point and check the whole manuscript before re-submission.

6. PLOS authors have the option to publish the peer review history of their article (what does this mean?). If published, this will include your full peer review and any attached files.

Reviewer #1: **Yes: **Amna Khan

Reviewer #2: No

---

## [Author Response · Author response to Decision Letter 0]

16 Dec 2020

Dear Christophe Hano,

1. We changed manuscript to Plos One style.

2. We changed the Funding Statement and Competing Interests Statement.

Funding Statement

“Although the author is employed by Nippon Soda and receives an annual salary, the company did not provide financial assistance or funding, and only provided equipment and facilities. Nippon Soda did not play a role in the study design, data collection and analysis, decision to publish, or preparation of the manuscript.”

Competing Interests Statement

“The authors have declared that no competing interests exist.”

Reviewer 1

•The title of research “effects of different lighting sources” is very general, rephrase it.

We revised title:

The effect of using light emitting diodes and fluorescent lamps as different light sources in growth inhibition tests of green alga, diatom, and cyanobacteria

•Please add an introductory line about why algal growth inhibition test is performed, discuss rationale behind the research.

We added sentence (LINE 13, 14):

Aquatic organisms have been used to investigate the safety of chemicals worldwide. One such assessment is an algal growth inhibition test.

•Line 44, 45, please elaborate if LED or FL is better? This literature does not support or negate the research study.

We added sentence (LINE 48-53):

LEDs have also been used for illumination in greenhouse plants. Singh D et al. (2015) said “It is necessary to further investigate the not yet fully understood physiological processes mediating plant responses to LED light; however, economic analysis has clearly shown that LEDs can reduce the electricity cost, and investment (high capital cost) will be returned as profit in long-term operations in greenhouse industries” [5]. Reports about the effect of LEDs on plant growth have led to several studies comparing LEDs with FLs.

We added our thought in the conclusion: LEDs thus appear advantageous over FLs in algal growth inhibition tests.

•Line 51, “algal growth inhibition tests” are cultured or algae are cultured?

We revised sentence (LINE 59, 60):

Currently, algae in algal growth inhibition tests are commonly cultured using growth chambers with FLs.

•Line 57-58, “as no comparison test between LED and FL was conducted” please rephrase it with “has been conducted”.

We revised sentence (LINE 66).

•Please mention about the role of LED/FL illumination in algal growth inhibition tests.

We added sentence (LINE 67, 68):

Light is essential for the growth of algae, and light sources are therefore important for algal growth inhibition tests.

•However, the figures added could be of better resolution.

We revised figure resolution.

•Please improve the discussion section, and make your research work more attractive.

We added sentence (Line 263-270).

Moreover, LEDs are being used increasingly in the commercial field where it is shown that the initial high capital cost of LEDs is returned as profit in certain sectors due to numerous advantages over FLs. Furthermore, LEDs do not contain mercury and have a low health impact due to low UV. LEDs have benefits in terms of decreased electricity cost and the reduced negative impact on the health of environmental organisms (including humans). In addition, they do not differ from conventional lighting sources when promoting growth of plants (including algae). LEDs thus appear advantageous over FLs in algal growth inhibition tests.

•Line 213, this information is not present in the cited reference 10. Please, check.

References:

We revised the reference on Line 213. We added a reference, therefore the reference number did not change.

•The references are not uniformly cited, e.g. reference 10, journal name is abbreviated while rest are not.

We revised reference style.

Reviewer 2

1. We revised title:

The effect of using light emitting diodes and fluorescent lamps as different light sources in growth inhibition tests of green alga, diatom, and cyanobacteria

2. We added sentence (LINE 71-74).

We selected P. subcapitata, N. pelliculosa, and A. flos-aquae because these algae are used to evaluate the ecotoxicity of industrial chemicals and pesticides and are recommended in test guidelines.

3. We added sentence (LINE 86-88).

P. subcapitata, A. flos-aquae and N. pelliculosa were used for this study because these algae are recommended in OECD test guideline No.201, OCSPP number 850.4500 and 850.4550. 

4. This manuscript was checked by native check service before the resubmission.

---

## [Decision Letter · Decision Letter 1]

5 Jan 2021

PONE-D-20-28966R1

The effect of using light emitting diodes and fluorescent lamps as different light sources in growth inhibition tests of green alga, diatom, and cyanobacteria

PLOS ONE

Dear Dr. Okamoto,

Thank you for submitting your manuscript to PLOS ONE. After careful consideration, we feel that it has merit but does not fully meet PLOS ONE’s publication criteria as it currently stands. Therefore, we invite you to submit a revised version of the manuscript that addresses the points raised during the review process.

We look forward to receiving your revised manuscript.

Kind regards,

Christophe Hano

Academic Editor

PLOS ONE

Reviewers' comments:

Reviewer's Responses to Questions

**Comments to the Author**

1. If the authors have adequately addressed your comments raised in a previous round of review and you feel that this manuscript is now acceptable for publication, you may indicate that here to bypass the “Comments to the Author” section, enter your conflict of interest statement in the “Confidential to Editor” section, and submit your "Accept" recommendation.

Reviewer #1: All comments have been addressed

Reviewer #2: All comments have been addressed

2. Is the manuscript technically sound, and do the data support the conclusions?

Reviewer #1: Yes

Reviewer #2: Partly

3. Has the statistical analysis been performed appropriately and rigorously? 

Reviewer #1: Yes

Reviewer #2: Yes

4. Have the authors made all data underlying the findings in their manuscript fully available?

Reviewer #1: Yes

Reviewer #2: Yes

5. Is the manuscript presented in an intelligible fashion and written in standard English?

Reviewer #1: Yes

Reviewer #2: Yes

6. Review Comments to the Author

Reviewer #1: (No Response)

Reviewer #2: Comment and suggestion to authors:

PONE-D-20-28966

1) The author addressed all the comments, but some answers are not so reasonable to be accepted in PLOS ONE. For example, the previous comment 3: “3) Why did Pseudokirchneriella subcapitata, Navicula pelliculosa, and Anabaena flos-aquae is suitable for this study? Why don’t the authors used other species or more number of species for their experiment? “

The authors responded that “This point should be clarified in the manuscript as well. We added sentence (LINE 86-88). P. subcapitata, A. flos-aquae and N. pelliculosa were used for this study because these algae are recommended in OECD test guideline No.201, OCSPP number 850.4500 and 850.4550. We could not conduct algal growth inhibition tests with other species, because we cannot buy fluorescent lamp.“

It can be understood, if the authors could not conduct algal growth inhibition tests with other species. But the acceptable reason should not be “we cannot buy fluorescent lamp”. An important criterion for the research that will be published in PLOS ONE is “Experiments, statistics, and other analyses are performed to a high technical standard and are described in sufficient detail.”

2) There are some spelling mistakes and grammatical error remained in this manuscript, the author should pay more attention on this point and check the whole manuscript before re-submission.

For example, in the punctuation marks in introduction section

….. “phosphor-converted (pc)-LEDs and FLs have similar emission spectra, so 63 pc-LEDs may be suitable as an FL replacement within the same color temperature range 64 without causing significant changes in algal growth rates and biochemical properties [10]…..

The authors started to use quotation marks without ending it.

7. PLOS authors have the option to publish the peer review history of their article (what does this mean?). If published, this will include your full peer review and any attached files.

Reviewer #1: **Yes: **Amna Khan

Reviewer #2: No

---

## [Author Response · Author response to Decision Letter 1]

14 Jan 2021

I appreciate your kind confirmation our manuscript during your busy time. We agree with you and revised our manuscript.

1) We revised LINE 86–89.

“P. subcapitata, N. pelliculosa, and A. flos-aquae are recommended in OECD test guideline No. 201 and OCSPP number 850.4500 and 850.4550 [6–8], and the test methods are internationally recognized. Moreover, the results of these species are required in registrations of pesticides or industrial chemicals. Therefore, these three algal species were used for this study.”

2) This manuscript was checked and revised by native speaker of English before the re-submission.

---

## [Decision Letter · Decision Letter 2]

8 Feb 2021

The effect of using light emitting diodes and fluorescent lamps as different light sources in growth inhibition tests of green alga, diatom, and cyanobacteria

PONE-D-20-28966R2

Dear Dr. Okamoto,

We’re pleased to inform you that your manuscript has been judged scientifically suitable for publication and will be formally accepted for publication once it meets all outstanding technical requirements.

Kind regards,

Christophe Hano

Academic Editor

PLOS ONE

Additional Editor Comments (optional):

Reviewers' comments:

Reviewer's Responses to Questions

**Comments to the Author**

1. If the authors have adequately addressed your comments raised in a previous round of review and you feel that this manuscript is now acceptable for publication, you may indicate that here to bypass the “Comments to the Author” section, enter your conflict of interest statement in the “Confidential to Editor” section, and submit your "Accept" recommendation.

Reviewer #2: All comments have been addressed

2. Is the manuscript technically sound, and do the data support the conclusions?

Reviewer #2: Yes

3. Has the statistical analysis been performed appropriately and rigorously? 

Reviewer #2: Yes

4. Have the authors made all data underlying the findings in their manuscript fully available?

Reviewer #2: Yes

5. Is the manuscript presented in an intelligible fashion and written in standard English?

Reviewer #2: Yes

6. Review Comments to the Author

Reviewer #2: Thanks for the revised version, the manuscript can be accepted to publish this time. However, some minor grammatical errors are remained in this manuscript.

7. PLOS authors have the option to publish the peer review history of their article (what does this mean?). If published, this will include your full peer review and any attached files.

Reviewer #2: No

---

## [Editor Report · Acceptance letter]

10 Feb 2021

PONE-D-20-28966R2 

The effect of using light emitting diodes and fluorescent lamps as different light sources in growth inhibition tests of green alga, diatom, and cyanobacteria 

Dear Dr. Okamoto:

I'm pleased to inform you that your manuscript has been deemed suitable for publication in PLOS ONE. Congratulations! Your manuscript is now with our production department. 

Kind regards, 

on behalf of

Dr. Christophe Hano 

Academic Editor

PLOS ONE